# Previous School Bullying-Associated Depression in Chinese College Students: The Mediation of Personality

**DOI:** 10.3390/bs13010014

**Published:** 2022-12-23

**Authors:** Hongjie Li, Xueyan Qian, Jian Gong, Haiying Dong, Xuejiao Chai, Hong Chao, Xiaolei Yang

**Affiliations:** 1School of Public Health, Qiqihar Medical University, Qiqihar 161000, China; 2Party Committee Office, Qiqihar Medical University, Qiqihar 161000, China; 3School of Pathology, Qiqihar Medical University, Qiqihar 161000, China

**Keywords:** previous school bullying, depression, Big Five personality traits, mediating

## Abstract

Previous school bullying was associated with increased risk of depression in students. However, little was known about the role of the Big Five personality traits in this association. The purpose of this study was to investigate the possible mediation by the Big Five personality traits in this association in a large group of Chinese college students, and to provide help for educators to prevent students from serious psychological and mental diseases caused by school bullying. Random stratified cluster sampling was used to survey 2152 college students ranging from freshmen to seniors at three universities in Qiqihar city, Heilongjiang Province, China. The risk factors for previous school bullying included gender, living expenses per month, caregivers, parents often quarreling, and divorced parents. Males were more likely to be bullied at school than females. The influencing factors of depression include gender, caregivers, living expenses per month, frequent parents quarreling, and parental divorce. Females were more prone to depression than males. Depression was significantly correlated with all dimensions of school bullying and the Big Five personality traits (*p* < 0.05). The Big Five personality traits were found to play a significant mediating role between depression and school bullying in up to 45% of cases involving depression. Our major findings highlighted the promising role of personality-based intervention measures in reducing the risk of depression associated with school bullying in Chinese students.

## 1. Introduction

School bullying is a worldwide phenomenon and one of the main challenges for the education community. At present, bullying in schools is common in many countries and regions. A European multicenter study reported a bullying victimization rate of 18.2% among children [1]. A Global School-based Health Survey showed that 20.6% of Indonesian students and 28.3 to 51.0% of Southeast Asian countries aged 13–17 years were bullied [2]. Previous studies also showed that 32% of students were bullied in some form by their peers at school on one or more days over the period of a month. Bullying and cyberbullying behaviors represent nowadays a major social problem, affecting 37% of adolescents [3]. Like in any other countries, school bullying was prevalent among students in China: cross-sectional studies revealed that the self-reported prevalence of traditional bullying victimization can be as high as 22.7%. In addition, the estimated prevalence was 20.8% for face-to-face bullying victimization, while 9.6% for cyberbullying victimization [4].

School bullying refers to any unwanted, repeated, and harmful aggressive behavior among children and adolescents. It can encompass four possible dimensions, including physical bullying (e.g., kicking, hitting, pushing, grabbing and destroying objects), verbal bullying (e.g., name-calling, teasing, abusing, mocking), relationship bullying (e.g., excluding from social situations, spreading rumors) and cyberbullying (e.g., threating through phones, text messages, WeChat, online emails, personal websites or online forums to attack others) [5,6,7]. Previous studies have shown that bullying victimization not only leads to increased behavioral and emotional problems [8], psychiatric symptoms [9], and reduced academic performance [10], but was also associated with increased risk of suicidal ideation and behaviors [11]. In addition, there is evidence that bullying is strongly associated with depression symptoms from adolescence to young adulthood [12]. Therefore, it is imperative to reduce the risk of depression associated with school bullying victimization in students. However, direct intervention on school bullying victimization is likely to be ineffective. Victims of bullying may choose to hide their experiences for fear of rejection by peers [13]. Under these circumstances, exploring intervenable factors in the association between school bullying victimization and depression among students will provide useful information. Previous studies suggested that the personality was a risk profile for bullying. It reported that bullies showed low agreeableness and conscientiousness, high extraversion and neuroticism, and, unexpectedly, also lower openness [14]. Nasti’s models showed a significant negative association between bullying and openness, conscientiousness and agreeableness [15]. Tani et al. indicated that children with low conscientiousness tend to act in antisocial ways [16]. Personality traits describe individual characteristics such as cognitive, emotional, and behavioral aspects that may play a role in diatheses or an increased propensity to psychopathologic states [17,18], including depressive symptoms. The Big Five personality traits define the five dimensions of personality such as neuroticism, extraversion, openness to experience, agreeableness, and conscientiousness. Different types of personality may also be associated with variable reactivities, such as emotional regulation or coping styles [17,19]. It was previously reported that neuroticism was a strong mediator in the association between gender and depression in addition to various social and psychological factors [20]. However, the role of Big Five personality traits in the occurrence of bullying and depression has not severely been explored among the Chinese population. Under this scenario, exploring intervenable factors in the association between school bullying victimization and depression among Chinese college students will provide useful information.

This study hypothesizes that the Big Five personality traits would have a mediating effect on the relationship between school bullying and depression. It aimed to analyze the influence of the Big Five personality traits on depression in students who had been bullied, and to provide reference data to help develop targeted psychological education to aid in the prevention of bullying. We tested three specific aims in the present study. First, we examined associations among each of the five personality factors, bullying, and depressive symptoms. Second, we explored the direct contribution rate of depressive symptoms after being bullied in school. Third, we examined the mediating roles of each personality factor in the relationship between bullying and depressive symptoms, meaning the indirect contribution rate of the Big Five personality traits generated after being bullied to the production of depressive symptoms.

## 2. Materials and Methods

### 2.1. Participants and Ethics Statement

This study was a cross-sectional survey conducted between September and October 2020 in Qiqihar city, Heilongjiang Province, China. A two-stage simple random cluster sampling method with probability proportionate to sample size design was used: in the first stage, 3 schools (a comprehensive university, a medical university, and a tertiary institution) were randomly selected from all schools in Qiqihar; in the second stage, according to the required sample size, 7–8 classes were randomly selected from each of the chosen schools, and all eligible students within the chosen classes were included. The paper version of the questionnaire was sent to the monitor of each class by the instructor of each class, and then sent and taken back from the students after the answer by the monitor. The paper version of the questionnaire was distributed to 2271 students from the three targeted schools. A total of 2152 (94.8%) responses were completed and returned with no missing items. These were included in this study, whereas the other 119 responses had missing values and were excluded. Of the sample population, 974 were males (45.3%) and 1178 were females (54.7%). The participants’ average age was 18.9 ± 1.62.

Approval for the study was obtained from the Ethics Committee of Qiqihar Medical University (2021-31). All the participants provided informed consent to participate in the study.

### 2.2. Measurements

#### 2.2.1. Demographic Characteristics and Measurement of Bullying

Demographic information regarding gender, place of residence, grades, academic years, cost of living, and caregivers were obtained in this study. Responses to whether their parents quarreled often or were divorced were dichotomized as either yes or no. The past bullying on campus was measured by the self-determined bullying items, such as the following three items: whether you were bullied in primary school, whether you were bullied in secondary school, and whether you were bullied in university. If bullying had occurred across at least one of three school stages, the respondent was deemed to have suffered from bullying.

#### 2.2.2. Measurement of Depression

Depression was measured using the Self-rating Depression Scale (SDS), which was developed by Zung in 1965 [21]. The SDS contains 20 items and its design was based on the diagnostic criteria for depression. Subjects rate each item with regard to the feelings they have experienced during the past several days using a 4-point Likert scale. The questionnaire included items such as “I feel down-hearted and blue”, “Morning is when I feel the best”, etc. The raw sum score of the SDS ranges from 20 to 80 but results are usually presented as the SDS Index, which is obtained by expressing the raw score converted to a 100-point scale. A total of 53–62 points is considered to indicate mild depression, 63–72 points indicate moderate depression, and 72 or more indicate major depression. Total scores on the SDS do not correspond with a clinical diagnosis of depression but rather indicate the level of depressive symptoms that may be of clinical relevance. It has been established as a valid, reliable instrument in several studies in order to measure depressive symptoms [22,23]. In the present study, the Cronbach’s alpha coefficient was 0.82.

#### 2.2.3. Measurement of the Big Five Personality Traits

The Big Five personality traits were measured using the Big Five Inventory (BFI) developed by John and Srivastava [24]. The BFI measures five personality traits: extraversion, agreeableness, conscientiousness, neuroticism, and openness. The inventory consisted of 44 items, and the extraversion dimension was measured by 8 items, the agreeableness dimension was measured by 9 items, the conscientiousness was measured by 9 items, the neuroticism dimension was measured by 8 items and the openness dimension was measured by 10 items. The measure was scored using a 5-point Likert scale (1 = Strongly disagree to 5 = Strongly agree). The questionnaire included items such as “I often feel scared”, “Once I have set my goal, I will stick to it”, “I think most people are basically kind-hearted”, etc. It has demonstrated good reliability and validity across many sample groups [25]. In the present study, the Cronbach’s alpha coefficient was 0.86.

### 2.3. Statistical Analysis Method

The dataset was analyzed using SPSS version 24.0 (IBM Corp., Armonk, NY, USA). For demographic data, chi-squared tests were used to analyze the categorical variables. Descriptive statistics concerning demographic and bullying information, personality traits, and psychological variables were presented in terms of number (*N*) and percentage (*%*) as appropriate. Moreover, the differences between school bullying and depression detected among different groups were compared by chi-square test or Wilcoxon rank sum test. Pearson’s correlation was used to examine correlations among experiences of being bullied, the Big Five personality traits, and depression. A mediation model was established and analyzed using hierarchical regression; in Step 2, bullying was entered; in Step 3, the Big Five personality traits were added. Standardized estimates of *β, R^2^*, and *R^2^*-changes (Δ*R^2^*) for each step were provided. The level of significance was set at 0.05 (two-sided).

## 3. Results

### 3.1. Comparison of School Bullying and Depression among Different Demographics

The demographic characteristics of college students, the distribution and difference analysis of their experiences of being bullied and depression (in categorical variables) are shown in Table 1. The risk factors for previous school bullying include gender, living expenses per month, caregiver, parents often quarreling, and divorced parents (*p* < 0.05). Boys were more likely to be bullied at school than girls (*χ^2^* = 95.49, *p* < 0.01). The influencing factors of depression include gender, caregiver, living expenses per month, parents often quarreling, and parental divorce (*p* < 0.05). Girls were more prone to depression than boys (*χ^2^* = −3.55, *p* < 0.01).

### 3.2. The Effect of having Experienced Bullying on Depression

There were significant differences in the depression of students after being bullied in all campuses (*Z* = −10.824, *p* = 0.001). Students who had been previously bullied on campus now have an overall detection rate of 27% of depression. The incidence of depression among students who have suffered bullying in elementary school, secondary school, and university was 18.8%, 17.5%, and 7.4%, respectively, as shown in Table 2.

### 3.3. Correlations among Experiences of Bullying, the Big Five Personality Traits, and Depression

The correlations among experiences of bullying, the Big Five personality traits, and depression are shown in Table 3. The four traits of conscientiousness, agreeableness, openness, and extraversion were all negatively related to depression and experiences of bullying; however, neuroticism was positively related to depression and school bullying.

### 3.4. Direct and Indirect Prediction Effects

A path analysis prediction model was built with bullying as the independent variable and depression as the dependent variable. The model showed that bullying had a direct positive predictive effect on depression (*B* = 6.036, *p* < 0.01), as shown in Figure 1. The model found that there was a direct relationship between school bullying and depression, and the personality traits of neuroticism, conscientiousness, agreeableness, openness, and extraversion also had significant indirect mediating effect between campus bullying and depression (*p*< 0.01). The analytical results revealed that the largest indirect associations via neuroticism were 1.67 (calculated as 2.780 × 0.599), as shown in Figure 2. Therefore, the hypothesis that the Big Five personality traits would play a mediating role between depression and having been bullied at school was verified. The direct effect of interaction between campus bullying and depression was 6.7%. When the mediating effect of the Big Five personality traits was included, previous school bullying was indirectly associated with the occurrence of depression in 51.5% of cases. An increase of 44.8% over that found concerning the direct effect, as shown in Table 4.

## 4. Discussion

In recent years, school bullying has become increasingly recognized as a major social issue, with research focusing more on this topic. Our study found that males were more likely to be bullied at school than females, but females were more prone to depression than males, same as some previous studies [26,27]. Males in adolescence have rapid physical development and strong physical strength, but their psychological development is still immature, competitive and impulsive with emotional ups and downs. Compared with females, it is easier for males to use violent means to solve problems in life, and thus to bully by insulting, beating or threatening persecution [28]. Females are more mentally sensitive than males, are more susceptible to external environmental influences, and are prone to emotional disorders, such as depression [29]. Adolescents living in a family environment where their parents often quarrel or single-parent families are emotionally indifferent and psychologically inferior due to the lack of care and education of one parent. Those bring not only material and affection deficits to young people, but more serious psychological and spiritual damage. They are unwilling to have more contact with other classmates in school, experience a lack of friends, and are easily affected emotionally and in terms of personality [30]. Therefore, it is easy for this group to experience bullying on campus and negative emotions. Our survey found that there were significant differences in campus bullying and depression experience for research subjects depending on caregivers; the same goes for living expenses per month. Due to the rapid economic development, a large number of rural laborers go out to work to change their living conditions in China instead of taking their children with them, resulting in many left-behind children. Some evidence has indicated that left-behind children have many psychological problems due to the parent–child separation, poor living environment, and lack of effective education and supervision [31]. In order to attract parents’ attention and concern, left-behind children deliberately engage in antisocial behavior, which leads to bullying on campus, or lack of parents’ company for a long time and low economic conditions may lead to children’s inferiority complex and depression. Therefore, perhaps increasing parents’ company and family economic status can reduce the occurrence of campus bullying and depression, which can be used to prevent the occurrence of campus bullying and depression.

The occurrence of school bullying seriously affects students’ study and life, resulting in negative emotional states, leading to depression, fearfulness, and even suicide [5]. Our study showed that students who had been bullied were more likely to suffer from moderate to severe depression, with a significant correlation between previous school bullying and depression scores, which indicated that school bullying was a risk factor for depression. These findings are consistent with those of Kaltiala-Heino et al. [32] and Lutrick et al. [33], where it was found that the higher the frequency of bullying, the more often there were cases of depression. Therefore, along with seeking to prevent the occurrence of school bullying, it is also necessary to pay close attention to psychological changes likely to be occurring among the victims of bullying and strengthen the relevant psychological aspects of education among students. Further, attention must be paid to improving student social skills, interpersonal skills, psychological robustness, and the use of appropriate skills when confronted with bullying. In addition, some studies have shown that there was a great relationship between personality and bullying on campus, which could be used as an important research field in the prevention of bullying [15]. Our findings are similar to those of the above researches. Consequently, the prevention of bullying on campus may be considered from the perspective of development of a stable personality in childhood and adolescence. This requires parents and teachers to pay attention to the daily changes of bad emotions and psychology in children and guide them.

The depression and bullying displayed a positive correlation with neuroticism and a negative correlation with conscientiousness, openness, agreeableness, extraversion. These findings fit the historic notion that neurotic introverts are especially vulnerable to depression [34]. Personality factors are known to be involved in the regulation of emotion and cognitive vulnerability that contribute to a predisposition to depression. Neuroticism is associated with negative emotions such as anxiety, fear, and anger [35]. Extraversion is the tendency to be active and sociable [36]. Openness to experience is the tendency toward preferring unconventional ideas and experiencing diverse emotions [37]. Agreeableness refers to interpersonal characteristics such as altruistic and cooperative tendencies. Conscientiousness is characterized by persistence, organization, and goal-directed behavior [38]. Bullying was a specific type of interpersonal violence. Victims of bullying usually reported stronger negative emotions and higher levels of emotion dysregulation compared with those who were not bullied [39]. It has been documented that lower levels of personality functioning in depressed patients were associated with a broader spectrum of negative emotions [40]. This was probably because individuals with a higher level of personality have more promoting and compensatory factors, and therefore may experience fewer symptoms of depression even if being bullied. Meanwhile, we determined that personality had an importantly intermediate association between school bullying and depression. Among all dimensions of personality, neuroticism played the strongest mediating role. Neuroticism had a significant positive predictive effect on developing depression, while conscientiousness, agreeableness, openness, and extraversion had a significant negative predictive effect, similar to the findings of previous studies [41]. This major finding highlighted the significance of personality in interventions designed to prevent depression associated with school bullying in Chinese students. Therefore, the incidence of depression caused by school bullying could be reduced through seeking to adapt students’ personality traits. Specifically, tendencies towards neuroticism, which leads to poor emotional stability and tends to orient behavior towards impulsivity, irritability, and psychological stress, need to be confronted and students should try their best to develop the ability to control their own emotions and form a stable personality. After the appropriate development of these character traits, students would be more likely to exhibit greater compassion and a sense of responsibility, engage in positive communication and establish affirming relationships with others, have good resilience and be optimistic concerning the future. They would also not be inclined to arbitrarily abuse, fight with, mock others, or spread malicious rumors, and, when dealing with instances of bullying, they would have appropriate psychological and emotional stability to enable them to confront bullying effectively, addressing negative emotions. Studies have also shown that individuals with extroverted, cheerful, easy-going, and self-disciplined personalities have better mental health, while individuals with greater levels of emotional instability or neurotic tendencies are more likely to have more psychological problems [42].

There are several strengths in the present study, including study sample size and methodological approaches. Although the relationship between previous school bullying and the occurrence of depressive symptoms were earlier reported across different countries, our study is the first to be performed in the China population examining the mediating roles of five factor personality traits in the link between bullying and depressive symptoms. The large sample size derived from a population cohort was useful for studying the risk factors and etiology that may contribute to the development of depressive disorders in Chinese college students.

Despite these strengths, there are also several limitations. First, this was a cross-sectional study, therefore causal inference should be avoided. Second, the retrospective self-report method used in collecting data may be prone to information bias. Finally, study participants were chosen from a specific province in northeast China, and representativeness should be considered when interpreting the results.

## 5. Conclusions

In conclusion, we determined that school bullying was significantly associated with depression; more importantly, personality traits may be related with depressive symptoms through direct or indirect paths via their influence on school bullying, especially for the dimensions of emotional regulation, family support, and interpersonal assistance. Our major findings highlighted the promising role of personality intervention measures in preventing depression associated with school bullying in children and adolescents. Future longitudinal studies are needed to corroborate our study results. In addition, our study was among the first attempts to discuss the mediating effect of personality in school bullying and depression in Chinese population. In the future, our research will continue to explore the influence of other mediators, such as gender, on the associated between campus bullying and depression.

## Figures and Tables

**Figure 1 behavsci-13-00014-f001:**
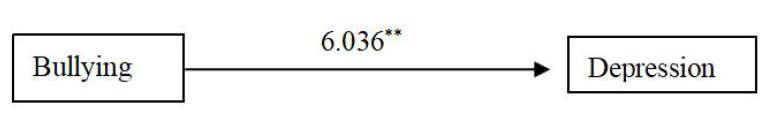
The pathway of predicting depression by school bullying. Note: ** *p* < 0.01, the value on the arrow indicates the B value of the regression coefficient.

**Figure 2 behavsci-13-00014-f002:**
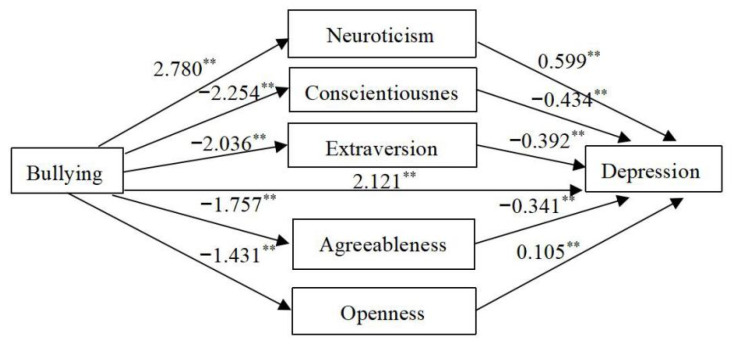
The path chart of predicting depression by school bullying and Big Five personality. Note: ** *p* < 0.01, the value on the arrow indicates the B value of the regression coefficient.

**Table 1 behavsci-13-00014-t001:** Descriptive statistical analysis of having experienced bullying and depression.

	Bullying (%)		Depression (%)				*χ^2^_b_/Z_b_*	*χ^2^_d_/Z_d_*
	No	Yes	No	Mild	Moderate	Severe		
Gender								
male	34.5	55.5	41.5	52.1	47.4	26.3	95.49 **	−3.55 **
female	65.5	44.5	58.5	47.9	52.6	73.7		
living expenses per month								
<$150	21.4	30.1	24.0	27.8	30.6	26.3	22.73 **	6.02 *
$150–300	72.3	63.1	69.5	65.1	64.1	63.2		
>$300	6.4	6.8	6.5	7.2	5.3	10.5		
Caregivers								
parents	88.6	80.7	84.9	83.6	76.1	73.7	15.76 **	14.02 **
grandparents	41.4	58.6	13.2	12.4	18.7	26.3		
relatives	1.3	2.3	1.3	3.0	1.4	0.0		
boarding	0.6	1.5	0.6	1.0	3.8	0.0		
Do parents often quarrel?								
yes	11.1	19.8	12.1	18.7	23.4	42.1	31.08 **	−5.57 **
no	88.9	80.2	87.9	81.3	76.6	57.9		
Have your parents divorced?								
yes	8.3	11.0	7.5	11.5	16.7	10.5	4.41 *	−4.40 **
no	91.7	89.0	92.5	88.5	83.3	89.5		

Note: b is bullying, d is depression, * *p* < 0.05, ** *p* < 0.01.

**Table 2 behavsci-13-00014-t002:** Analysis on the differences of categorical variables between bullying and depression.

The Period of Bullying	Depression (*n*/%)				*Z*	*p*
	No	Mild	Moderate	Severe		
Being bullied in primary school						
no	858 (39.9)	375 (17.4)	111 (5.2)	8 (0.4)	−13.41	<0.001
yes	396 (18.4)	295 (13.7)	98 (4.6)	11 (0.5)		
Being bullied in secondary school						
no	1052 (48.9)	402 (18.7)	108 (5.0)	12 (0.6)	−14.31	<0.001
yes	202 (9.4)	268 (12.5)	101 (4.7)	7 (0.3)		
Being bullied at university						
no	1200 (55.8)	574 (26.7)	152 (7.1)	12 (0.6)	−11.45	<0.001
yes	54 (2.5)	96 (4.5)	57 (2.6)	7 (0.3)		
Overall situation of previous bullying						
no	731 (34.0)	255 (11.8)	59 (2.7)	4 (0.2)	−10.82	0.001
yes	523 (24.3)	415 (19.3)	150 (7.0)	15 (0.7)		
Total	1254 (58.3)	670 (31.1)	209 (9.7)	19 (0.9)		

**Table 3 behavsci-13-00014-t003:** Correlation Analysis of Variables (*r*).

	1	2	3	4	5	6	7
1. Depression	1						
2. Neuroticism	0.343 **	1					
3. Conscientiousness	−0.553 **	0.061 **	1				
4. Agreeableness	−0.499 **	−0.049 *	0.546 **	1			
5. Openness	−0.501 **	0.143 **	0.773 **	0.534 **	1		
6. Extroversion	−0.372 **	−0.055 *	0.553 **	0.364 **	0.644 **	1	
7. Bullying	0.259 **	0.190 **	−0.141 **	−0.181 **	−0.102 **	−0.117 **	1

Note: * *p* < 0.05, ** *p* < 0.01.

**Table 4 behavsci-13-00014-t004:** The model explains the change.

Model	*R*	*R^2^*	Δ*R^2^*	Error in Standard Estimates
Direct effect	0.259	0.067	0.067	11.288
Mediation effect	0.719	0.517	0.515	8.137

## Data Availability

The data that support the findings of this study are available from the corresponding author upon reasonable request.

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
