# Peer review of "Previous School Bullying-Associated Depression in Chinese College Students: The Mediation of Personality"

_behavsci, 2022, doi:10.3390/bs13010014_

Round 1

Reviewer 1 Report

I realize that great work and time have been devoted to this paper. It has a lot of strengths, but I think that some changes should be recommended. 

Title: the title does not adequately reflect the content of the paper. Please, try to change it to better inform the readers about the relationships between the variables you test and also inform them about the quality of your sample.

Introduction

The literature revision has some references that are too old. Besides citing some papers from 2001, you can consider some relevant papers on the topic RECENTLY published in other Journals. There are some Journals that suggest a high percentage of references published during the last five years. The introduction is too brief. The most relevant findings related to the constructs under study should be summarized.

Methodology

The Instruments or Questionnaires section needs more information. Please, some examples of items should be provided to the readers. If you can, please inform me about previous studies where the same instrument has been used and the reliability obtained in that research.

Discussion:

First of all, try to adjust your conclusions to the findings better. Or to say in other words, please try to justify more clearly the connection between your conclusions and your findings.

Finally, a section related to limitations, future lines of investigation, and the principal contributions of the research could be attractive. Your paper has a lot of relevant implications for educators, psychologists, society, and policymakers, but you need to elaborate more on this topic.

Reviewer 2 Report

see attached file

Reviewer 3 Report

The manuscript reports on a correlational study of demographics and school bullying and depression, including the mediating role of the Big Five personality traits. Strengths include a relatively large sample size and adequate methodology. The analysis and conclusions mostly acknowledge the correlational nature of the study, but sometimes the authors write as if causal relations were shown – for instance, that bullying causes depression. Further, the study could be put into a richer context of earlier research. Finally, the discussion includes some subjective and unsupported claims, which need attention. However, I see the weaknesses possible to address in writing since the study, data, and analysis have sufficient quality.  

Detailed comments, section by section:

Abstract

The names of the universities do not add much to the abstract.

Statistical details (p and chi-2 values) are usually not given in the abstract.

The unique contribution of the study, in relation to the context of earlier research, could be clarified in the abstract.

Introduction

The introduction very briefly introduces the topic and motivates the research question. It could be expanded to place the current study in a richer context.

“School bullying is a worldwide phenomenon and a growing but relatively new and challenging research area.” (lines 27-28). Further description of this relatively new research area could be given, along with references, to clearer show the relation between the current study to the research area.

“has never been explored among the Chinese population” (lines 30-31). If it has been explored in other populations, these studies should be reported.

Experiences of being bullied not only may cause harm to the body but may also lead to changes in personality and behavior”: (lines 38-39) This could be supported by references.

There is ample evidence…” (lines 39-42) could give more than one reference consider the quote.

Materials and Methods

Was the survey distributed on paper or online?

Is there reason to expect a systematic loss (e.g., the least or perhaps the most bullied students did not respond)?

Was the order of parts of the survey controlled? Is there reason to believe that the order may influence how the respondents answered?

How was the survey introduced to the respondents – what were the instructions?

Results

Table 1: The presentation of the depression results is a bit unclear: e.g., for boys it seems the numbers 41.5 represent no depression, 52.1 mild, 47.4 moderate, and 26.3 severe, respectively.

“after being bullied” (lines 125): How do you know that depression came after being bullied, and not before?

“the path analysis model showed that having experienced bullying played a direct positive role in the development of depression” (lines 161-162) and “Previous bullying at school had a direct effect on the occurrence of depression” (line 173): These conclusion lacks support, since the analysis only shows “predictive effect” (that is, correlation rather than causation).

The strengths of relationships and associations are sometimes reported, but also discussing them could make the conclusion more nuanced – e.g., gender effects on bullying and depression.

Discussion

There are several very speculative, and seemingly culturally dependent, conclusions drawn (e.g., lines 185-190). These could be more carefully marked as speculations or proposed explanations and given more support through references.  

The unique contribution of the current study should be clarified (although it is touched upon at the very end of the paper).

Writing: check usages of “extraversion” vs. “extroversion” throughout the paper.

Round 2

Reviewer 2 Report

The article has now been improved considerably.  However the authors still do not review earlier research on big five personality, and bullying.  This includes Tani et al. (Mentioned before) and Mitsopoulou 2015  (https://doi.org/10.1016/j.avb.2015.01.007).  

This should be reviewed in the Introduction, and findings compared in discussion (an opportunity to discuss cultural issues perhaps, as well).

Reviewer 3 Report

I thank the authors for the revisions made.

Most of the issues I raised have been solved by the authors.

However, in lines 248-275, there are still many statements that are subjective, over-generalizing and seemingly based on "cultural wisdom". These need to be removed or reformulated with sufficient support to scientific literature.
